# Energy Utilization Potential of Wheat Straw in an Ecological Balance—A Case Study of Henan Province in China

**Lyu Zhichen** * **and Zhu Xuantong**

Department of Quantitative and Technical Economics, Graduate School of Chinese Academy of Social Sciences, Beijing 102488, China; juliazxt@163.com

\* Correspondence: lvzchen@163.com; Tel.: +0086-18875095316

**Abstract:** Minimum volume of straw should be retained in collecting wheat straw in order to obtain sustainable agricultural biomass energy and measure the energy utilization potential of wheat straw when considering ecological balance. Based on relevant literature, this paper calculates the minimum, medium, and maximum volume of wheat straw retention in various types of soil and designs three different scenarios of minimum, medium, and maximum wheat straw retention. Taking Henan province in China as a case, this paper calculates the potential of wheat straw for energy utilization using linear regression method and scenario analysis, with consideration of influencing factors such as the harvest coefficient and combustion ratio of wheat straw. The results show that the energy utilization potential of wheat straw in Henan province in 2020, 2030, 2050 are 13.77, 16.48, 22.54 million tons of coal equivalent (TCEs), respectively, in the minimum retention scenario, assuming that wheat straw is not directly used for combustion. Excessive straw left in the field causes resource waste and produces $CH_4$ and other greenhouse gases. This paper finds that energy potential of wheat straw for energy utilization is limited when ecological balance is considered, however, it is beneficial to the sustainable development of crop biomass energy.

**Keywords:** biomass energy; wheat straw retention; ecological balance; China

## 1. Introduction

Ecological balance means that within a certain period of time, the organisms in the ecosystem and the environment, as well as the various species of organisms, achieve a highly adaptive, coordinated, and unified state through energy flow, material circulation, and information transmission [1]. The ecological balance in the paper refers to the realization of soil balance and sustainable development of resources. As a fundamental economic driving force and key for environmental protection, energy needs to be adequately supplied. Renewable energy, including biomass energy, provides a solid foundation for promoting social development and progress. According to the IEA's (International Energy Agency) Renewables 2018 market analysis and forecast report [2]: Renewables will continue their expansion in the next five years, covering 40% of global energy consumption growth; modern bioenergy is the overlooked giant of the renewable energy field, and its share in the world's total renewables consumption is about 50% today. China leads global growth in renewable energy as a result of policies to decarbonize all sectors and reduce harmful local air pollution, and becomes the largest consumer of renewable energy, surpassing the European Union by 2023. By 2023, the share of renewable energy in global energy demand is expected to increase by one-fifth to 12.4%. According to the BP (British Petroleum) Statistical Review of World Energy 2018 [3], world biofuels production increased by 3.5% in 2017, well below the 10-year average of 11.4%, but the fastest for three years. By fuel

type, global ethanol production grew at a similar rate of 3.3%, contributing over 60% to total biofuels growth. Biodiesel production rose by 4%, driven mainly by growth in Argentina, Brazil, and Spain. The development and utilization of crop straw in China has an important impact on agricultural ecology, especially on arable land soil. With the speeding up of low-carbon energy transition and expansion of climate change awareness, the demand for renewable energy is growing continuously. As a major component of renewable energy, biomass energy has a great significance not only for energy transition, but also for agricultural ecological balance in the environment. Therefore, in view of the current situation of international energy security, energy consumption, and environment, China needs to change the mode of energy consumption. Developing and utilizing renewable energy rationally is an important topic for China. In order to develop biomass energy while keeping ecological balance in China, China has to study how to evaluate, develop, and utilize crop residues on condition of maintaining the agricultural ecological environment.

As one of the top agricultural provinces with superior geographical location, Henan province produces a large volume of crop surplus and agricultural byproducts in China (as shown in Figure 1). According to the National Statistics [4], in 2016, its crop-sowing area reached 14.47 million hectares, accounting for 8.7% of the total crop sowing area in China, and ranking first place in China. Its grain output accounts for 9% in China. Its wheat-planting area is 5.47 million hectares (accounting for 37.8% of crop-planting area in Henan province and ranked first place) in Henan province. Henan province has a huge production of wheat straw, and enjoys the advantage of easy collection due to planting concentration. Therefore, this paper's discussion on the biomass energy potential of wheat straw in Henan province will provide reference for research in China's major crop biomass and new ideas for study on energy transition. This paper studies sustainable development and utilization of straw biomass energy in the typical area of Henan province in China, providing guidance for the advancement of biomass energy development in China under potential ecological and economic circumstances.

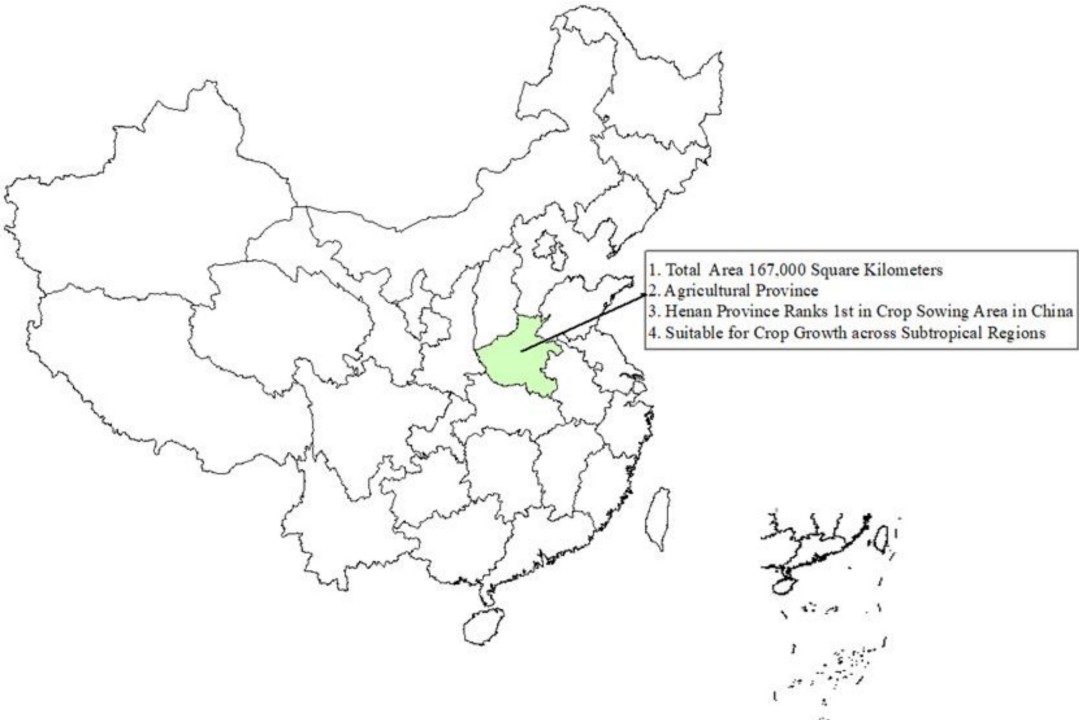

1. Total Area 167,000 Square Kilometers
2. Agricultural Province
3. Henan Province Ranks 1st in Crop Sowing Area in China
4. Suitable for Crop Growth across Subtropical Regions

**Figure 1.** Location of Henan province in China.

Since crop straw retention volume has an important impact on the soil function of cultivated land, previous studies mainly focused on maintaining adequate volume of straw for the purpose of soil protection. For example, studies are either conducted from the perspectives of preventing soil

erosion and wind erosion and maintaining the balance of soil organic matter, or the prediction of biomass energy potential and the sustainability of energy supply based on various models. In terms of preventing soil erosion from damaging soil function, RUSLE (Revised Universal Soil Loss Equation) is the main model adopted [5–10]. For example, when considering the influence of soil vegetation on soil erosion, Wang et al. evaluated the soil erosivity in a typical thin-layer black soil area of a northeast Pennsylvania river basin by using the erosion–productivity impact model, general soil erosion equation, and modified soil erosion equation [9]. In terms of maintaining the balance of soil organic matter, Wilhelm et al. discussed the harvesting height of corn straw in some regions of the United States based on soil type and planting mode, and suggested that the harvesting height of corn straw was between 40% and 55% of the vegetation height [11]. Sainju et al. studied the effect of different crop cultivation methods on soil organic matter [12]. In order to maintain the balance of soil organic matter, Manyowa et al. discussed the harvest threshold of crops [13]. Wang et al. discussed the way of straw mulching, the volume of straw mulching, and the degree of influence of fertilizer on soil organic matter [14]. Sonja et al. made a comprehensive evaluation of the potential of biomass energy development in some European countries by establishing relevant evaluation models [15]. Cai et al. calculated the straw energy potential that can be used as biomass energy by taking several influencing factors into account [16]. Sun et al. elaborated the application of biomass energy in fuel ethanol, biodiesel, biomass power generation, biogas, biomass liquefaction, materials, and other aspects [17]. Muth et al. evaluated the agricultural residues and calculated the volume of agricultural residues as biomass energy [18]. Zheng et al. studied straw mulching and straw decomposition, evaluated the influencing factors of sustainable agricultural development, and analyzed the benefits of straw mulching and its advantages for sustainable agricultural development [19]. Under the background of the national biofuel targets, Adusumilli et al. analyzed the use of some marginal land development for more agricultural biomass energy, and by using the Soil Water Assessment Tool (SWAT), evaluated the intensive pasture and pointed out that U.S. national energy policy designed around achieving energy independence should also consider environmental and economic trade-offs for biofuel to be an economically and environmentally sustainable alternative to fossil fuels [20]. Piqin Gong et al. adopted the trinomial tree model of real options to estimate renewable power projects and calculated the benchmark prices [21]. Miglietta et al. analyzed the water impact resulting from the use of various types of crops in the production of biodiesel from a quantitative point of view [22]. Miglietta et al. provided a first glance at addressing optimal resource allocation and the design of biodiesel imports in the EU regulation context, contributing to reducing the water footprint, while considering the economic aspects [23].

　　The literature mentioned above studied the current situation of soil function and straw distribution and utilization. However, there is limited literature on exploring the energy potential of wheat straw by designing different scenarios of soil straw retention. The straw retention proportion has a direct influence on the harvest straw quantity, which has an indirect influence on the development of biomass energy. Based on the research above, combined with the current situation of wheat straw utilization, this paper evaluates the biomass energy potential of wheat straw by designing different scenarios from the perspective of ecological balance. In addition, wheat straw has a variety of uses, such as for feed or industrial use (e.g., paper-making), including those left in the field to maintain soil functions; the wheat straw studied in this paper can be collected except for the straw employed for these uses. This paper considers the influence of straw retention volume on ecological balance and combining with different soil types, taking Henan province as a case. This paper studied wheat straw retention volume under three different scenarios of minimum (a), medium (b), and maximum (c) straw retention volume in Henan province. The energy potential of wheat straw in Henan province in three different scenarios (2020, 2030, and 2050) were calculated based on various factors such as the proportion of wheat straw used for feed, availability rate, and combustion substitution rate.

## 2. Materials and Methods

### *2.1. Calculation Process of Energy Utilization of Wheat Straw Biomass Based on Ecological Balance*

Based on the sown areas of wheat, soil types, and the area share of different soil types over the years in Henan province, the wheat straw yield was calculated by regression analysis. Based on relevant literature, the minimum straw retention of different soil types was obtained by referring to the research data, and three scenarios of minimum (a), medium (b), and maximum (c) straw retention were designed. According to different scenarios, the minimum retention demand per unit area of soil in the corresponding areas in Henan province was calculated based on different soil types, the minimum volume of straw retention of the corresponding types, and results from relevant literature. Considering other uses of wheat straw (such as the proportion of wheat straw used for feed and combustion) and the unit yield of wheat, the biomass energy potential of biomass energy of wheat straw was calculated. The direct combustion of straw pollutes the environment and its efficiency of energy utilization is low. As a result, the straw used for burning in the future may be replaced by other energy sources with the improvement of science and technology. Taking the combustion substitution proportion together, the benchmark scenario of no substitution for direct combustion, the medium scenario of 50% substitution for direct combustion, and 100% substitution for direct combustion were designed. The calculation process is shown in Figure 2. According to the results from literature in the field, straw retention was designed in three different scenarios, which are minimum retention, medium retention, and maximum retention, to determine the minimum straw retention on different types of soil.

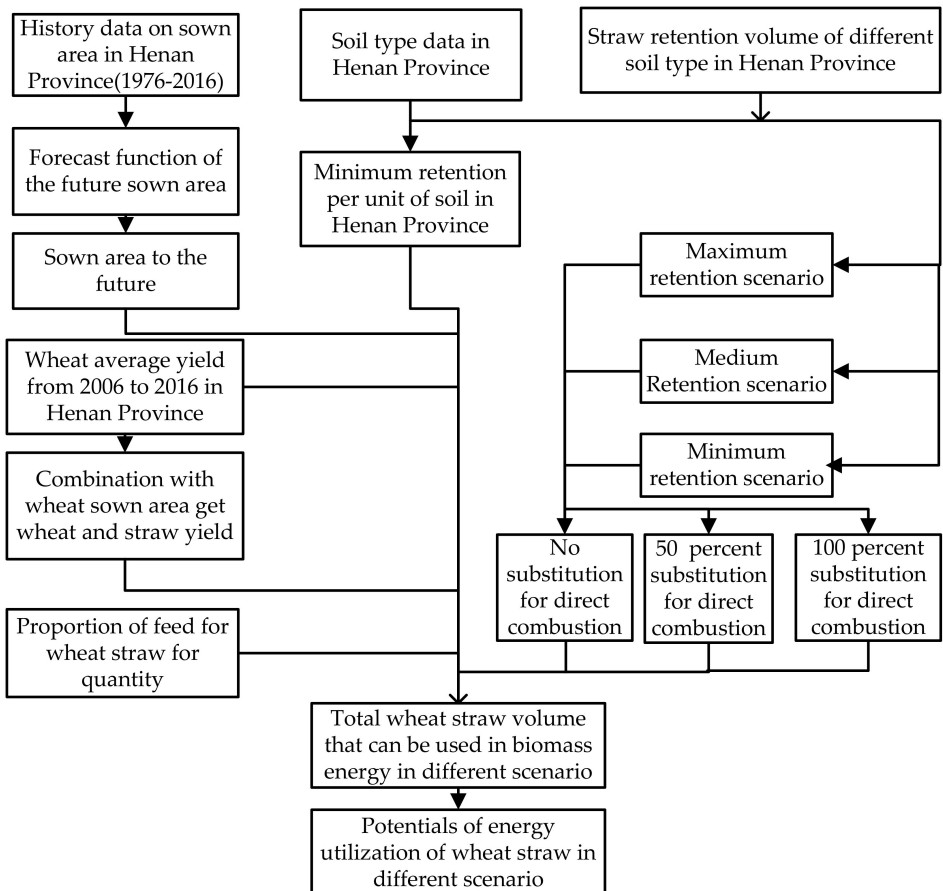

**Figure 2.** Calculation process of biomass energy utilization of wheat straw based on ecological balance in Henan province.

### 2.2. Prediction of Wheat-Sown Area and Yield in Henan Province

According to the National Data (1976–2016) [24], the data of wheat-sown area and unit yield in Henan province from 1976 to 2011 show that the changing tendency of wheat yield per unit is not significant. From the prediction of future sown area of wheat in Henan province, the sown area of wheat presents a linear growth. Based on a variety of crops growing demand and the change of the total crops sown area from National Data [24] (1976–2011), the wheat-sown area will eventually come close to a critical volume. Taking reference of the growing trend of corn-sown area over the years, wheat-sown area will gradually increase to 7000 ha in the coming decades. Extrapolating from the trend of the 1976–2016 data, the wheat-sown area will be 5.64, 6.02, and 6.78 million hectares in 2020, 2030, and 2050, respectively. Thus the increase will be about 3%, 10%, and 24%, respectively (compared to 5.47 million hectares in 2016). Taking into account the land area constraint, the average wheat-sown area in 2020, 2030, and 2050 is designed at 6.14 million hectares based on linear regression prediction.

Regarding the prediction of wheat yield per unit area, according to the China statistical yearbook [24], the wheat output per unit area from 1978 to 2016 increased by 108 kg/hm$^2$ (Figure 3). With reference to Lu's [25] prediction of China's medium-term crops such as situational volume (2020 and 2030), Chen's [26] design of the main crops' long-term production situational volume (2030 and 2050), and National Data, this paper predicts that compared with 2016, the wheat output per unit area will increase by 6% in 2020, 24% in 2030, and 57% in 2050, reaching 6749.58kg/ha, 7832.28kg/ha, and 9997.68 kg/ha, respectively.

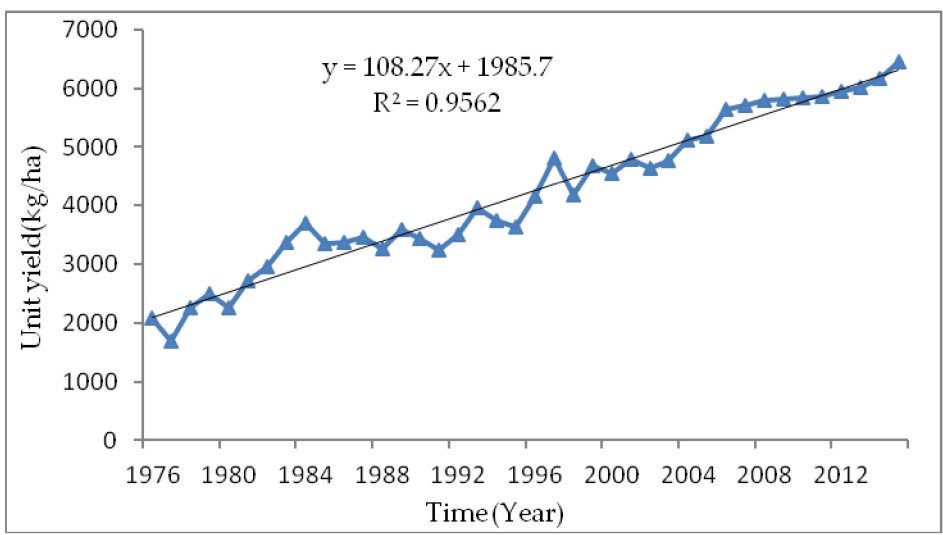

**Figure 3.** Output per hectare of wheat from 1976 to 2016.

### 2.3. Calculation Method of Wheat Straw Yield

Currently there is no statistical data on wheat straw yield in China. Relevant literature shows that wheat straw yield is usually calculated based on the straw harvest coefficient (the ratio between grass and grain). As the straw harvest coefficient will be influenced by different factors (such as climate, geography, cropping systems, collection method, measuring method), the crop straw harvest coefficients according to Kim et al. [27], Zeng et al. [28], Yang et al. [29], and Cai et al. [16] are 1.30, 1.34, 0.73, and 1.16, respectively. According to the data from Wilhelm [9], algorithm of soybean straw from Liu et al. [30], a comprehensive literature study, and adjustment to the actual situation of wheat growth, this paper designs a linear regression model of wheat straw yield ($y_{straw}$) and wheat yield ($y_{wheat}$):

$$y_{straw} = 0.6 y_{wheat} + 2.5 \qquad (1)$$

Here, $y_{straw}$ represents wheat straw yield (unit: t/hm$^2$), $y_{wheat}$ represents wheat production (unit: t/hm$^2$).

## 2.4. Calculation Method of Wheat Straw Retention Considering Ecological Balance

According to the results of literature review and the scenario simulation designed by Liu et al. [30] and taking the ecological balance into consideration, straw retention was designed in three different scenarios of minimum retention (a), medium retention (b), and maximum retention (c) to determine the minimum retention of straw on different soil types.

(1) In the scenario of maximum retention (c), given a certain soil type, the maximum volume of the minimum retention volume of straw to maintain soil function was determined according to the soil type corresponding to the relevant literature. According to the statistics of literature data, the soil types of wheat growing in Henan province are mainly divided into clay, loam, and sand.

$$\begin{cases} M_{\mathrm{max},1} = \max(M_{1,1}, M_{1,2}, \dots M_{1,m}) \\ M_{\mathrm{max},2} = \max(M_{2,1}, M_{2,2}, \dots M_{2,n}) \\ M_{\mathrm{max},3} = \max(M_{3,1}, M_{3,2}, \dots M_{3,h}) \end{cases} \tag{2}$$

In the equation shown, $M_{\mathrm{max},1}$, $M_{\mathrm{max},2}$, and $M_{\mathrm{max},3}$ represent the minimum straw retention volume of clay, loam, and sand under the condition of maximum retention, respectively. $M_{i,j}$ represents the minimum retention of wheat straw in $i$th soil in the $j$th literature.

(2) In the scenario of medium retention (b), the medium volume of the minimum retention volume of straw to maintain soil function was determined according to the soil type corresponding to the relevant literature. According to the statistics of literature data, the soil types of wheat growing in Henan province are mainly divided into clay, loam, and sand.

$$\begin{cases} M_{\mathrm{med},1} = \mathrm{med}(M_{1,1}, M_{1,2}, \dots M_{1,m}) \\ M_{\mathrm{med},2} = \mathrm{med}(M_{2,1}, M_{2,2}, \dots M_{2,n}) \\ M_{\mathrm{med},3} = \mathrm{med}(M_{3,1}, M_{3,2}, \dots M_{3,h}) \end{cases} \tag{3}$$

In the equation above, $M_{\mathrm{med},1}$, $M_{\mathrm{med},2}$, and $M_{\mathrm{med},3}$ represent the minimum straw retention volume of clay, loam, and sand under the condition of medium retention, respectively. $M_{i,j}$ represents the minimum retention of wheat straw of the $i$th soil in the $j$th literature.

(3) In the scenario of minimum retention (a), the minimum volume of the minimum retention volume of straw to maintain soil function was determined according to the soil type corresponding to the relevant literature. According to the statistics of literature data, the soil types of wheat growing in Henan province are mainly divided into clay, loam, and sand.

$$\begin{cases} M_{\mathrm{min},1} = \min(M_{1,1}, M_{1,2}, \dots M_{1,m}) \\ M_{\mathrm{min},2} = \min(M_{2,1}, M_{2,2}, \dots M_{2,n}) \\ M_{\mathrm{min},3} = \min(M_{3,1}, M_{3,2}, \dots M_{3,h}) \end{cases} \tag{4}$$

In the equation above, $M_{\mathrm{min},1}$, $M_{\mathrm{min},\,2}$ and $M_{\mathrm{min},3}$ respectively represent the minimum straw retention volume of clay, loam, and sand under the condition of minimum retention. $M_{i,j}$ represents the minimum retention of wheat straw in the $i$th soil in the $j$th literature.

## 2.5. Calculation Method of Crop Yield of Wheat Straw Considering Ecological Balance

Referring to the feed proportion ($\alpha$) and fuel proportion ($\beta$) set ($\alpha$ = 0.15, $\beta$ = 0.2) in the research literature of Wang et al. [31], Y is the harvest volume of straw:

$$Y = (1\alpha)(1\beta)\left(\mathrm{y}_{straw}\mathrm{w}_s\sum_{i=1}^{3}(\delta_i M_i)\right) \tag{5}$$

Here, $\mathrm{y}_{straw}$ represents the harvest volume of wheat straw (million tons), $\mathrm{w}_s$ represents the sown area of wheat ($10^3$ ha), in which $\delta_i$ refers to the proportion of loam ($i$ = 1,2,3) in total soil in Henan

province, and $M_i$ refers to the minimum volume of straw retention in class $i$ soil [32]. According to the data from China National Soil Survey Office [33], the soil types and sown area are sorted out.

## 3. Results

### 3.1. Wheat Yield and Sown Area

According to the calculation of Section 2.2 in this paper, the average sown area of wheat in Henan province in 2020, 2030, and 2050 (on the basis of the upper limit constraint) is 6.14 million hectares. As the total output of wheat = sown area of wheat * unit yield of wheat, the yield is obtained according to the predicted unit yield of wheat and the wheat straw yield is calculated from formula (1). Under the assumptions above, the sown area of wheat remains unchanged. According to the predicted future yield per unit area of wheat, the total output of wheat in 2020, 2030, and 2050 will be 38.08, 47.13, and 67.75 million tons, respectively. The total output of wheat straw will be 36.95, 43.318, and 57.60 million tons in 2020, 2030, and 2050 correspondingly.

### 3.2. Soil Equilibrium Scenario Considering Ecological Balance

By referring to relevant literature [11–13,34–37], minimum wheat straw retention volume was obtained in different scenarios and different literature, with different literature on soil types. As shown in Table 1:

**Table 1.** Minimum volume of straw for different soil.

| Soil Type | Scenario Design | Minimum Retention Volume (ton/ha) | Reference List |
|---|---|---|---|
| | Minimum | 0.95 | [13] |
| Loam Soil | Medium | 1.68 | [37] |
| | Maximum | 3.65 | [34] |
| | Minimum | 0.57 | [12] |
| Clay Soil | Medium | 1.93 | [34] |
| | Maximum | 4.86 | [35] |
| | Minimum | 0.66 | [37] |
| Sandy Soil | Medium | 1.78 | [11] |
| | Maximum | 3.25 | [36] |

### 3.3. Wheat Straw Demand Considering Soil Equilibrium

In consideration of ecological balance, it is necessary to maintain a balanced demand for straw. According to soil records of Henan province [33], soil types and distribution areas of different soil types are obtained. The basic process is as follows:

(1) According to the soil records of Henan province, various types of soil areas were obtained, and with Liu's [30] calculation method of soybean straw, the soil proportion distribution was set up.

(2) The wheat yield was obtained by referring to National Data [24].

(3) The minimum and maximum volume of the minimum retention volume of different soil types were obtained through literature review, and the median volume were calculated.

(4) According to relevant literature, by using formulas (2), (3), and (4), and referring to Liu's design on the soil straw retention volume model [32], this paper obtained the maximum, median, and minimum volume of the minimum retention volume per unit area in Henan province.

(5) With the count factors of formula (5), the soil records from Henan province, and the design of main soil types in Henan province, the loam, clay, and sand of cultivated wheat land were 0.4, 0.4, and 0.2, respectively. Multiplied by minimum retention volume of straw in different scenarios corresponding to certain types of soil (Table 1), minimum retention volume of wheat straw per unit area of different types of soil in Henan province were obtained in various scenarios (as shown in Figure 4). Finally, the straw's minimum retention volumes, 0.7 t/ha, 1.8 t/ha, 4.05 t/ha, of soil in Henan province were calculated for minimum, medium, maximum retention scenarios, respectively.

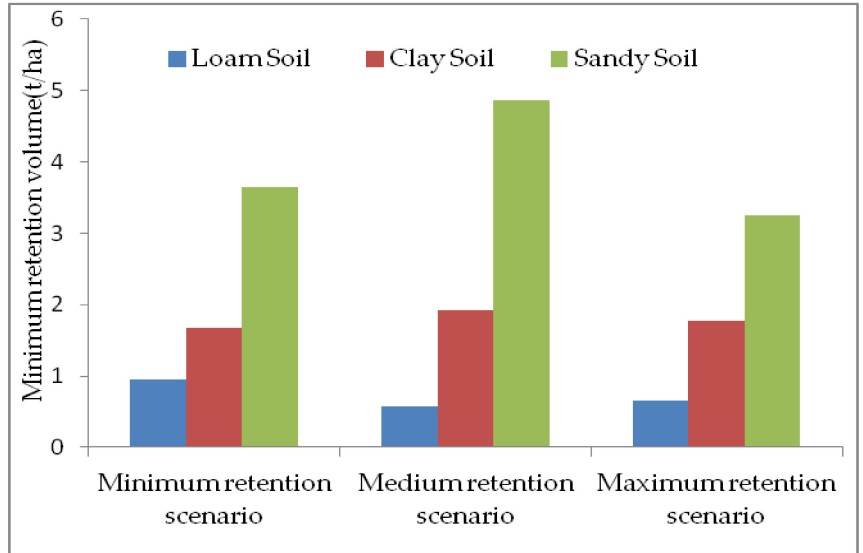

**Figure 4.** Minimum straw retention volume of soil for Henan in different scenarios.

### 3.4. Wheat Straw Yield for Biomass Energy

The evaluation of wheat straw yield is designed for minimum retention scenario (a), medium scenario (b), and maximum scenario (c). The degree of using straw for direct combustion as a substitute in different scenarios is designed in three scenarios: no substitution for direct combustion (the proportion of wheat straw for direct combustion remains unchanged), 50% substitution for direct combustion (the proportion of half of the wheat straw for direct combustion is replaced by other energy), and 100% substitution for direct combustion (the proportion of wheat straw for direct combustion is zero). The volume of wheat straw yield that can be collected in different scenarios is shown in Figure 5.

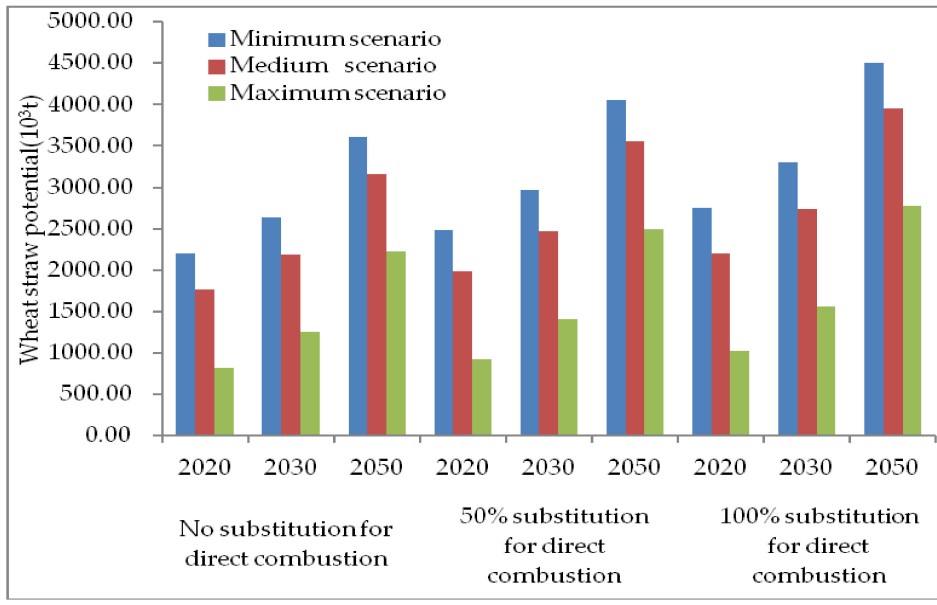

**Figure 5.** Potential of wheat straw in different substitution combustion scenarios. The volume of wheat straw that can be obtained in three scenarios with different combustion substitution rates.

(1) In the minimum scenario, wheat straw retention can be calculated by using formula (5); the harvest volume of wheat straw in Henan province in 2020, 2030, and 2050 are 22.03 million tons, 26.36 million tons, and 36.07 million tons, respectively. As the living standard of rural areas will continue to improve, the proportion of wheat straw for direct combustion as cooking fuel or heating fuel will

be reduced.①Assuming that there are other energy sources that can replace 50% of wheat straw for combustion, it can be calculated that wheat straw output in Henan province in 2020, 2030, and 2050 will be 24.79, 29.66, and 40.58 million tons, respectively.②Assuming that wheat straw will no longer be used for direct combustion, wheat straw output in Henan province in 2020, 2030, and 2050 will be 27.54, 32.95, and 45.09 million tons, respectively.

(2) Similarly, it can be seen that in the medium retention scenario: ① Assuming the proportion of wheat straw for direct combustion remains unchanged in the future, it can be predicted that the wheat straw output in Henan province in 2020, 2030, and 2050 will be 17.60, 21.93, and 31.64 million tons, respectively. ② In the scenario of 50% substitution for direct combustion, the wheat straw output in Henan province in 2020, 2030, and 2050 will be 19.81, 24.68, and 35.60 million tons, respectively. ③ In the scenario of 100% substitution for direct combustion, the wheat straw output in Henan province in 2020, 2030, and 2050 will be 22.00, 27.42, and 39.55 million tons, respectively.

(3) Similarly, it can be seen that in the maximum retention scenario: ① Assuming the proportion of wheat straw for direct combustion remains unchanged in the future, it can be predicted that the wheat straw output in Henan province in 2020, 2030, and 2050 will be 8.19, 12.52, and 22.22 million tons, respectively. ② In the scenario of 50% substitution for direct combustion, the wheat straw output in Henan province in 2020, 2030, and 2050 will be 9.21, 14.08, and 25.00 million tons, respectively. ③ In the scenario of 100% substitution for direct combustion, the wheat straw output in Henan province in 2020, 2030, and 2050 will be 10.23, 15.64, and 27.78 million tons, respectively.

## 3.5. Analysis of Potential Energy Utilization of Wheat Straw

Referring to the conversion coefficients calculated by other scholars by using energy conversion technology, also known as the conversion coefficient commonly used in China nowadays, this paper chooses 0.5 as the coefficient (referring to Liu's [38] design principles of calculation on China's biomass and distribution). The energy utilization potential of wheat straw in Henan province in 2020, 2030, and 2050 are estimated as shown in Figure 6. In the scenario of minimum retention and 100% substitution for direct combustion, the energy utilization potential of wheat straw in Henan province in 2020, 2030, and 2050 are 13.77, 16.48, and 22.54 million TCEs, respectively. In the scenario of maximum retention and zero substitution for direct combustion, the energy utilization potential of wheat straw in Henan province in 2020, 2030, and 2050 are 5.12, 7.82, and 13.89 million TCEs, respectively.

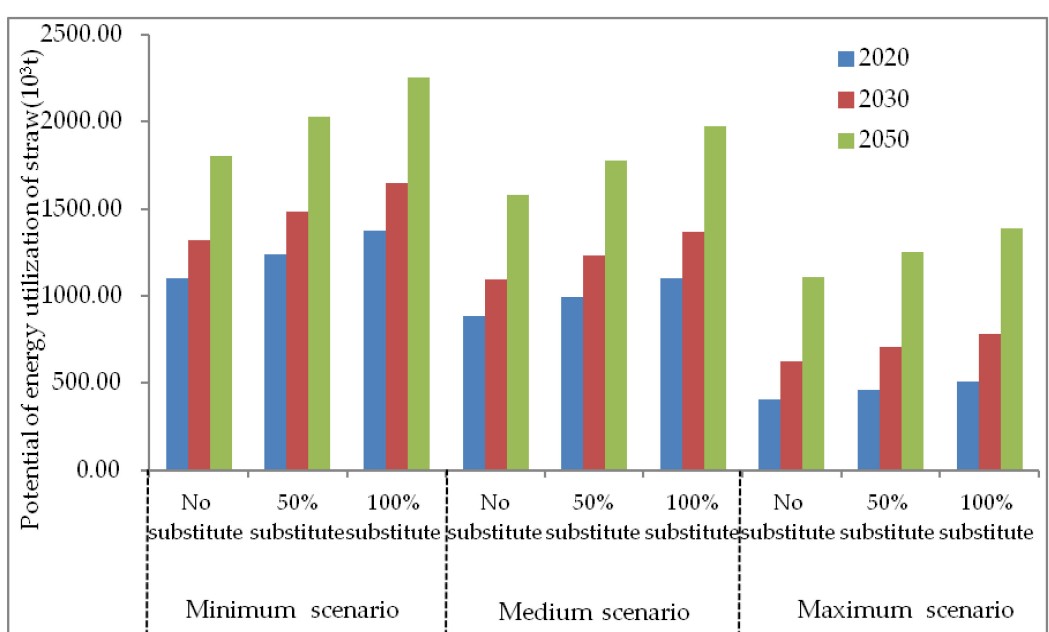

**Figure 6.** Biomass energy utilization potential of wheat straw in Henan Province.

## 4. Discussion

This paper chooses Henan province, a representative agricultural province in China, as the object of study, and studies minimum straw retention on three different types of soil in minimum, medium, and maximum scenarios. Then, by sorting the data of the area of loam, clay, and sand in Henan province, combining with the total planting area of wheat in Henan province, calculating the minimum retention volume of straw when considering ecological balance, and designing three different scenarios, this paper obtains the potential of wheat straw in Henan province for developing biomass energy in 2020, 2030, and 2050.

From the research results, when considering ecological balance, the biomass energy potential of wheat straw may decrease in the short term. However, as a whole, it is beneficial not only to agricultural development, but also to sustainable development of biomass energy and the ecosystem. The direct combustion of straw pollutes the environment and its efficiency of energy utilization is low. With continued improvement of living standard in rural China, wheat straw burning in rural areas will gradually decrease in the future and the volume of wheat straw that can be used for biomass energy will increase and hence the energy utilization potential of wheat straw will be improved. If wheat straw is no longer used for direct combustion in the future, in the medium retention scenario designed in this paper, energy utilization potential of wheat straw in Henan province will reach 11.03, 13.71, and 19.78 million TCEs in 2020, 2030, and 2050, respectively. The results provide reference for reasonable development of biomass energy of wheat straw in Henan province and lay a solid foundation for future study on potential of wheat straw for energy utilization in China.

There are spaces for further discussion in the following fields, for instance, how to identify and analyze the factors affecting the development of agricultural biomass energy and other renewable energy resources and evaluate the design of other relevant parameters. In further studies, scholars may explore other factors influencing the development of biomass energy and evaluate the potential of other agricultural biomass energies, so as to provide a theoretical basis for sustainable development of agricultural biomass energy.

## 5. Conclusions

This paper finds that considering ecological balance and soil balance, we need to design a reasonable straw retention volume. With the increasing demand for energy, environmental degradation, and rapid economic development, the development of agricultural biomass energy surely goes with the tide. According to the calculation, the potential of biomass energy in China is huge. In order to make more effective use of agricultural biomass energy, to promote healthy development of the biomass energy industry, and to increase the use of biomass products, the government should formulate and improve relevant laws and regulations on development and utilization of biomass energy. At the same time, the development of agricultural biomass energy in China is in line with the strategic direction of the world's renewable energy development. Excessive straw left in the field not only causes resource waste, but also produces $CH_4$ and other greenhouse gases. Therefore, straw retention should be taken into account during future agricultural biomass energy exploitation. Technological exchange, application, and deployment of biomass energy technologies need to be promoted in order to meet the challenges in an era of energy transition and climate change mitigation.

**Author Contributions:** Conceptualization, L.Z.; Formal analysis, L.Z.; Investigation, L. Z.; Methodology, L.Z.; Supervision, Z.X.; Visualization, Z.X.; Writing-original draft, L.Z.; Writing-review & editing, L.Z.

**Funding:** This paper was supported by The National Social Science Fund of China, grant number 18AJY009.

**Acknowledgments:** The authors would like to express their appreciation to the anonymous referees for their thoughtful suggestions for improving the quality of their paper. They would also like to thank May Zhang and Rabindra Nepal for their editing of their paper.

**Conflicts of Interest:** The authors declare no conflict of interest.

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
