# Peer review of "Energy Utilization Potential of Wheat Straw in an Ecological Balance—A Case Study of Henan Province in China"

_resources, doi:10.3390/resources8010041_

Round 1

Reviewer 1 Report

I remember that I already reviewed the former version of this paper, and I can observe that very few of the changes recommended have been accomplished. I appreciate a full update of the dataset, more recent than the previous one, but the paper still lacks of references, especially in the field of study exposed. The language is still poor, so a grammar review is also recommended.

In order to improve literature, I strongly recommend to check the following papers, which perfectly suits the subject treated:

Miglietta, P. P., De Leo, F., & Pizzi, S. (2016, March). The Water Footprint of Energy Crops for the Support of Italian Biodisel Production. In Proceedings of the XXVII Congresso Nazionale di Scienze Merceologiche-Qualità & Innovazione per una Economia Circolare ed un Futuro Sostenibile, Viterbo, Italy (pp. 2-4).

Miglietta, P. P., Giove, S., & Toma, P. (2018). An optimization framework for supporting decision making in biodiesel feedstock imports: Water footprint vs. import costs. Ecological Indicators85, 1231-1238.

Qinglong Shao & Lei Rao (2018) The rebound effect of dematerialization and decoupling: a case of energy efficiency, Chinese Journal of Population Resources and Environment.

Definitely, I recommend a major revision, again.

Author Response

Thank you very much for your suggestions and recommended references. According to your suggestions, we have revised the background section, including more information relevant to the current study. And we have improved the references and the English language used.

Reviewer 2 Report

Although the efforts for the revision, the paper has not improved. I am sorry but I suggest  to reject it.

Author Response

Thank you. We have improved the paper.

Reviewer 3 Report

Hello authors. I have some overall comments that I would like to see you address to make the paper stronger. 

Overall, the paper needs very extensive editing for English language, grammar, and style. I found some sections hard to follow, so having a trusted colleague who is well versed in English read your work would help. 

For the background section, particularly the first paragraph on page 3, it reads like a laundry list of papers you have read. By separating out every background study into its own sentence, I felt the paragraph was boring and did not add to the paper. Consider condensing and revising the background section. Including more information relevant to the current study, and even some policy implication sources, would be great. 

For the methods, I had a hard time distinguishing what, exactly, you did in terms of solving different scenarios. Perhaps this is a language barrier issue on one or both sides, but I'm not sure I follow the difference in the scenarios, as well as the difference between straw and wheat straw. It also seemed like equations 2-4 were the same essentially, and I do not follow how you made your calculations. Please be more clear on this. Are you trying to calculate projected future values of total wheat-straw availability? Are you looking to determine the potential value in BTU of wheat-straw? 

Big picture, how does the ecological economy play into your work? In the conclusion section you briefly mention policy implications, but you should consider fully addressing those concerns/ideas earlier in the paper. 

The figures were helpful, though they are fuzzy. Consider saving them in vector format (pdf, png) and then inserting into the work. Copy and paste makes them low-quality. But I think the bar graphs are useful and add value. 

You mention how damaging wheat-straw combustion is to the environment, but you are projecting increased wheat-straw use? Will this mean a bigger environmental impact? I am confused on what your overall message is, and how you incorporate combustion, crop yield, and environmental impact into a cohesive story. 

If you have land values for each year (2020, 2030, 2050), why did you use an average? Also, what is significant about those selected years? 2020 is next year, so wouldn't that match current values today in 2019?

You mention a growing trend for corn and wheat, are the two linked? Are all crops increasing in China?

Again, please extensively edit for English and readability. I think the topic is of interest, and could be useful for people to read. If you are able to rework the paper and address my comments above, I think the paper would be much improved and could potentially be ready to publish. But for now, please address my concerns. Good luck with your research!

Author Response

Thank you for your valuable advice. First of all, we have invited the experts who is well versed in English modify the grammar and style for our paper.

According to your suggestion, in the background section, particularly the first paragraph on page 3, we have revised the background section, including more information relevant to the current study, and even some policy implication sources.

For the methods (equations 2-4), we have described the calculation in detail in the article. As your understand, we are trying to calculate projected future values of total wheat-straw availability and looking to determine the potential value in BTU of wheat-straw.

The ecological balance in the paper refers to the realization of soil balance and sustainable development of resources. According to your suggestion, we have changed the term ecological economy to ecological balance.

The problem of the uploaded version led to the problem of the pictures. We will send the pictures and charts in the article (in vector format) to the editor separately by email.

The wheat straw designed in this paper is not directly burned after collection, but it is processed by relevant technologies to generate electricity, which will reduce the environmental impact caused by direct combustion and inefficient use.

About the relationship between wheat and corn, we design the article by use the results from the National Data, not all the crops are growing, but the wheat and corn sown area of the overall situation of change is more similar, and according to the trend extrapolation, measure the sown area of increased proportion in 2020, 2030 and 2050, but considering the land area of a cap, it is relatively reasonable to choose the average value for calculation.

Reviewer 4 Report

The articles is well conceived and this version - the paers has parts written in two colours what indicates probably that it integrates responses to other reviews- seems good, either in terms of content, structure, clairty, methods, results and also discussion. I think the quality of the presentation is also go, that it is sound in scientific terms and can be of interest for many readers. Of course it can be improved, for instance the references could be more up-to-dated and the English language used can be improved, too, with a carefull and pacefull revision.

Nevertheless I suggest publication of the paper after these small corrections.

Author Response

Thank you for your valuable comments and recognition. We have improved the references and the English language used. Thank you very much.

Round 2

Reviewer 1 Report

The paper achived an average rating in terms of overll merit. For this reason it caould be accepted for publication in RESOURCES

Author Response

Thank you very much.

Reviewer 2 Report

The authors improved the paper that can be published in the present form.

Author Response

Thank you very much.

Reviewer 3 Report

Hello Authors. Thank you for your response to my previous concerns. At this time, I have additional comments for you.

The paper still needs extensive English editing to be worthy of publication. There are still too many awkward sentences that disrupt the flow of your work. 

I still feel the introduction is too lengthy without adding much value. I would rather have more of an introduction to the problem and why your work is important, and not a list of all the past works that have ever been done. Please refine the introduction so that the reader understands the point of the study. 

Please make the distinction between wheat straw and what is being harvested/studied more clear. Your responses to my comments were helpful, and I would like to see similar explanations of the work provided in the paper. 

Author Response

Thank you for your valuable advice. First of all, we have invited two experts who are well versed in English modify the grammar and style for our paper.

According to your suggestion, we have revised and abbreviated the introduction. In addition, we have made the distinction between wheat straw and what is being harvested/studied in the paper. The wheat straw has a variety of uses, which can be used for feed or industrial uses (such as paper-making), including those left in the field to maintain soil functions, the wheat straw studied in this paper can be collected except for these uses.

Thank you very much.